# Treadmill exercise modulates the medial prefrontal-amygdala neural circuit to improve the resilience against chronic restraint stress

Zhihua Luo[1,6], Junlin Chen[1,6], Yelin Dai[1], Kwok-Fai So [1,2,3,4,5✉] & Li Zhang [1,3,4,5✉]

Aerobic exercise effectively ameliorates mental disorders including anxiety and depression. Current findings mainly attribute its neural mechanism to the improvement of adult neurogenesis, while leaving the possible circuitry mechanism unclear. In the current study, we identify the overexcitation of the medial prefrontal cortex (mPFC) to basolateral amygdala (BLA) pathway under chronic restraint stress (CRS), and 14-day treadmill exercise selectively reverses such abnormalities. Using chemogenetic approaches, we find that the mPFC-BLA circuit is necessary for preventing anxiety-like behaviors in CRS mice. These results collectively suggest a neural circuitry mechanism by which exercise training improves the resilience against environmental stress.

[1] Key Laboratory of CNS Regeneration (Ministry of Education), Guangdong-Hong Kong-Macau Institute of CNS Regeneration, Jinan University, Guangzhou, China. [2] State Key Laboratory of Brain and Cognitive Science, Li Ka Shing Faculty of Medicine, The University of Hong Kong, Hong Kong SAR, China. [3] Center for Brain Science and Brain-Inspired Intelligence, Guangdong-Hong Kong-Macao Greater Bay Area, Guangzhou, China. [4] Neuroscience and Neurorehabilitation Institute, University of Health and Rehabilitation Sciences, Qingdao, China. [5] Center for Exercise and Brain Science, School of Psychology, Shanghai University of Sport, Shanghai, China. [6] These authors contributed equally: Zhihua Luo, Junlin Chen. ✉email: hrmaskf@hku.hk; zhangli@jnu.edu.cn

Physical exercise is one effective way to decrease the risk of anxiety disorders as supported by independent lines of human studies[1–3] as well as rodent models[4–6]. The exploration of the biological mechanism of endurance training on mental health mainly focused on the potentiation of hippocampal neurogenesis[7,8], the relief of oxidative stress[9] or neuroinflammation[10], whilst little has been known about its neural circuitry mechanisms. In human cohorts, brain imaging studies have reported altered connectivity among anxiety-related brain regions including parietal cortex[11], orbitofrontal cortex and cingulate gyrus[12], caudate nucleus and ventral anterior putamen[13]. It is worth noted that the activity of amygdala nuclei, one critical region for anxiety behaviors, was markedly affected by repetitive exercise trainings[14]. These results add more insights for the change of brain network under exercise, although more mechanistic studies are required to elucidate the neural circuit.

Among different brain nuclei mediating anxiety behaviors, the basolateral amygdala (BLA) is one central hub as it receives inputs from cortical and subcortical nuclei to integrate sensory and mental status of the body[15]. Under chronic restraint stress (CRS), BLA presented dramatic changes of synaptic plasticity[16] and membrane conductance[17], suggesting the disruption of related brain networks to induce anxiety behaviors. Among input regions of BLA, the medial prefrontal cortex (mPFC) showed highly enriched interconnectivity, and this mPFC-BLA pathway is activated in the mouse CRS model[18]. Given a recent finding that exercise training in humans changed the connectively between mPFC and amygdala[19], we propose that mPFC-BLA circuit may play a role in exercise-improved resilience upon environmental stress.

To test this hypothesis, we generated a mouse CRS model, on which 14-day treadmill exercise effectively prevented the occurrence of anxiety-like behaviors. Ex vivo electrophysiological recordings demonstrated the hyperexcitation of the mPFC-BLA circuit in both presynaptic projecting neurons of mPFC and postsynaptic BLA neurons under CRS, and exercise training recovered neuronal activity to basal levels. Further chemogenetic manipulations demonstrated that the inhibition of mPFC-BLA pathway was necessary for exercise-mediated anxiolysis. These results suggested the neural circuitry mechanism of physical exercise in improving anxiety disorders.

## Results

**Exercise prevents stress-induced anxiety-like behaviors and inhibits BLA-projecting mPFC neurons.** We first generated a mouse CRS model by applying daily physical restraint stress on adult male mice for 14 consecutive days. The exercised group also received 1-hr daily treadmill exercise after the CRS paradigm (Fig. 1a). Behavioral assays found no change of general motor activity in the open field or the elevated plus-maze among all groups ($P > 0.05$; Fig. 1b, d). However, the duration in the central region or in the open arm was remarkably decreased in CRS group and was recovered with exercise training, while exercise in unstressed mice had no significant behavioral effect (Fig. 1c, e). These results agreed with our previous findings[20] and suggested the prevention of anxiety-like behaviors by physical exercise.

As one of neural substrates for anxiety-like behaviors, the activation of mPFC-BLA circuit occurs after CRS[18,21]. We thus studied the activity of this pathway by identifying the BLA-projecting mPFC neurons via injecting adeno-associated virus (AAV) vector with transsynaptic retrograde labeling (AAV-Retro-EGFP) into BLA (Fig. 1f). These BLA-projecting neurons were predominantly found in layer 5 the prelimbic region (PrL) of mPFC, where we performed the electrophysiological recording on acute brain slices (Fig. 1f). Given fixed depolarizing current,

CRS group presented higher frequency of spikes, which were depressed by exercise (Fig. 1g, h). On the contrary, the non-BLA projecting cells in PrL showed lower excitability under CRS, and presented more spikes by exercise training (Fig. 1i, j). These data implied the differential modulation of projecting neurons in PrL by exercise. Furthermore, we characterized the excitatory-inhibitory (E/I) balance of these 2 different groups of projecting neurons via the whole-cell patch recording. The quantification of miniature excitatory postsynaptic currents (mEPSCs) of BLA-projecting neurons found higher amplitudes and frequencies under CRS, and remarkable suppression of excitatory currents in exercise group (Fig. 1k–m). On the other hand, the inhibitory currents (mIPSCs) of BLA-projecting neurons were depressed by CRS and were elevated under exercise training, as supported by the amplitude and frequency (Fig. 1n–p). In non-BLA projecting PrL neurons, we observed differential regulations, as exercise training re-elevated mEPSCs (Fig. 1q–s) and suppressed mIPSCs (Fig. 1t–v). Interestingly, exercise training on naïve, unstressed mice did not change the excitability or E/I balance of either of these 2 groups of PrL neurons (Fig. 1g–v). These data collectively suggested the selective inhibition of mPFC-BLA circuit by exercise training on stressed mice.

**Physical exercise prevents anxiety via suppressing the mPFC-BLA circuit.** Having observed the anxiolytic effect and inhibition of mPFC-BLA pathway by exercise, we investigated if this neural circuit was sufficient for the improvement of stress resilience. The effect of exercise training was mimicked by expressing the inhibitory chemogeneics receptor hM4Di into BLA-projecting neurons under the direction of AAV-Retro-Cre vector within BLA (Fig. 2a, b). The infusion of receptor ligand clozapine N-oxide (CNO) remarkably attenuated the total spike number, suggesting decreased excitability upon chemogenetic manipulation (Fig. 2c, d). As functional evidence, chronic infusion of CNO alleviated anxiety-like behaviors in CRS group as shown by the recovery of normal exploration activity toward the central zone or the open arms (Fig. 2e, g), whilst the total distance remained unaffected (Fig. 2f, h). Therefore, the inhibition of mPFC-BLA pathway recapitulated the anxiolytic effect of exercise training.

Subsequently, we verified the necessity of the circuitry inhibition underlying exercise-mediated anxiolysis. Using a similar dual-viral strategy, we selectively expressed the excitatory chemogenetics receptor hM3Dq into BLA-projecting mPFC cells (Fig. 3a, b). The re-activation of these neurons by CNO effectively elevated their excitability (Fig. 3c, d). In behavioral assays, the chemogenetic activation in exercised mice did not affect the general motor activity (Fig. 3f, h) but markedly decreased the time duration in the central zone or in the open arms (Fig. 3e, g), indicating the relapse of anxiety-like behaviors. In sum, exercise training inhibits the mPFC-BLA neural circuit, conferring improved resilience against environmental stress.

**Physical exercise modifies the E/I balance of BLA neurons.** The manipulation of BLA-projecting mPFC cells effectively mimicked or abolished exercise effect in counteracting CRS-induced anxiety-like behaviors. We further described the anatomical identity of this pathway. An anterograde transsynaptic strategy in which an AAV2/1-Cre vector was injected into PrL, plus the expression of Cre-dependent fluorescent proteins in BLA nuclei, was combined with AAV-Retro-EGFP injection to label possibly bidirectional connection between PrL and BLA (Fig. 4a, b). Quantification showed <20% of cells co-expressing dual fluorescence in BLA (Fig. 4c), suggesting the specificity of anterograde labeling. Next, using $Ca^{2+}$-dependent calmodulin kinase II alpha (CaMKIIα) as the marker, most of BLA-projecting mPFC

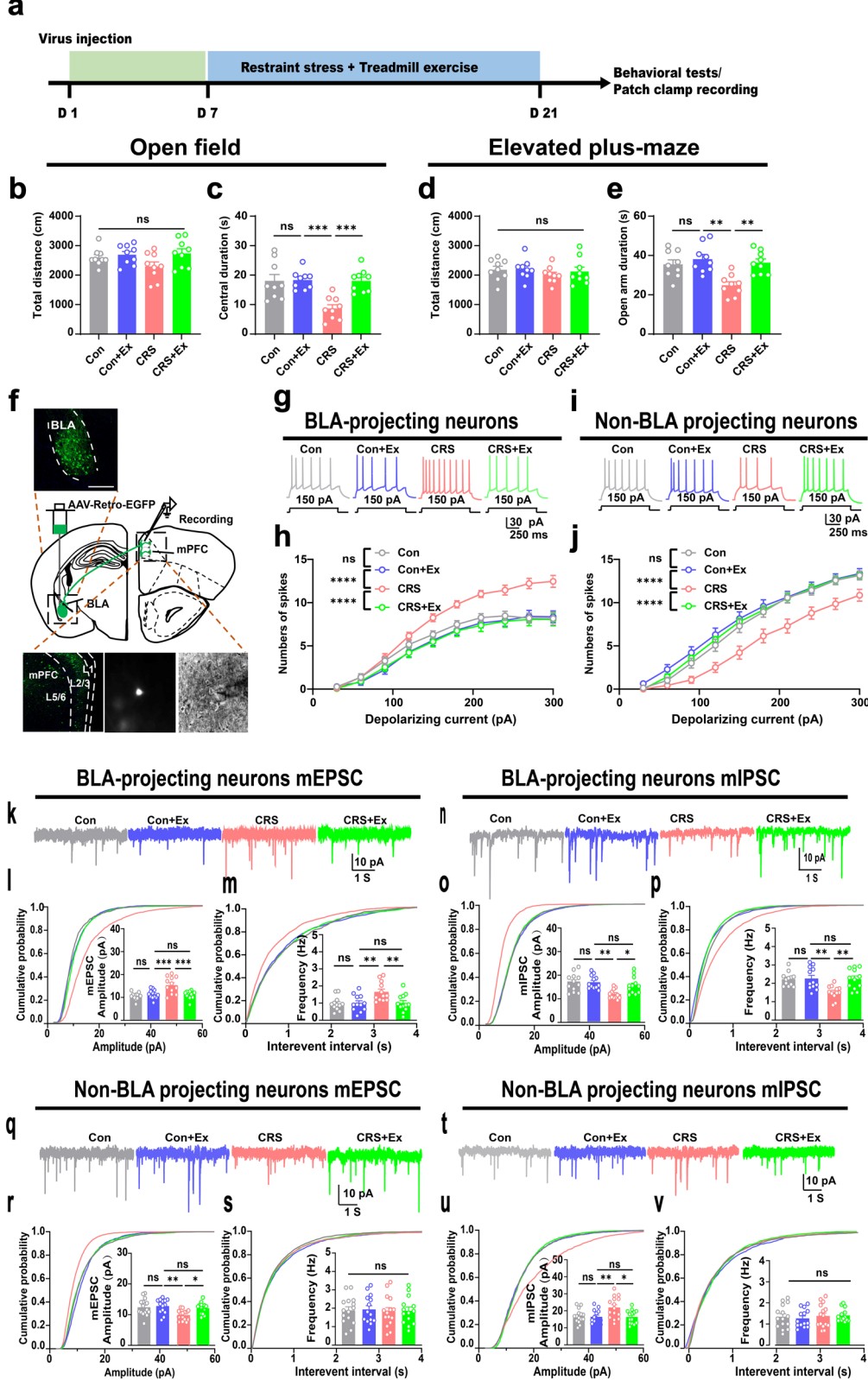

neurons were found to be glutamatergic cells (Fig. 4d–f). Since both pyramidal tract (PT) and intratelencephalic (IT) cells exist in the mPFC projecting neurons[22,23], we further injected retrograde label cholera toxin subunit beta (CTB) to the contralateral side of the AAV-Retro-EGFP injection (Fig. 4g). Immunofluorescent co-labeling showed that >70% of BLA-projecting cells belonged to IT subtype as they also innerved cortex on the contralateral side

(Fig. 4h, i). Alternatively, when using ipsilateral injection of CTB into the periaqueductal gray (PAG), we found that only a minor fraction of BLA-projecting cells belonged to the PT group (Fig. 4j–l). The collateral projection of mPFC neurons was further excluded by examining the spatial distribution of axonal terminus of BLA-projecting mPFC neurons (Fig. 4m, n). Last, the possible effect of exercise on VTA-projecting neurons was excluded as

**Fig. 1 Treadmill exercise inactivates BLA projecting neurons in mPFC. a** Time schedule of the experiment. **b** Total distance in the open field. One-way ANOVA, $F_{(3,32)} = 2.183$, $P = 0.1093$. **c** Central duration in the open field. $F_{(3,32)} = 9.739$, $P = 0.0001$. **d** Total distance in the elevated plus-maze. $F_{(3,32)} = 0.4678$, $P = 0.7068$. **e** Time duration in the open arm. $F_{(3,32)} = 7.947$, $P = 0.0004$. $N = 9$ mice in each group in (**b**–**e**). **f** Experimental schemes of retrograde labeling of BLA-projecting mPFC neurons. Upper panel, injection site of AAV-Retro-Cre in BLA. Lower panels, fluorescent images (left) and enlarged single neuron for cell attaching record (middle and right). Scale bar, 250 μm. **g** Representative traces of action potential under the fixed injection currents (150 pA). **h** Total spike number of BLA-projecting neurons under gradient depolarizing currents. Two-way ANOVA with respect to the group effect, $F_{(3,710)} = 50.84$, $P < 0.0001$. $n = 20$ neurons from 4 mice in each group. **i** Representative traces of action potential under the fixed injection currents (150 pA). **j** Total spike number of non-BLA projecting neurons under gradient depolarizing currents. $F_{(3,520)} = 50.97$, $P < 0.0001$. $n = 13$ neurons from 4 mice in each group. **k** Sample traces of miniature excitatory postsynaptic current (mEPSC). **l** Cumulative probability and comparison of means of mEPSC amplitude. One-way ANOVA, $F_{(3,44)} = 9.794$, $P < 0.0001$. **m** Cumulative probability and means of mEPSC frequency. $F_{(3,44)} = 5.958$, $P = 0.0017$. **n** Sample traces of miniature inhibitory postsynaptic current (mIPSC). **o** Cumulative probability and comparison of means of mIPSC amplitude. One-way ANOVA, $F_{(3,44)} = 6.429$, $P = 0.0010$. **p** Cumulative probability and means of mIPSC frequency. $F_{(3,44)} = 6.388$, $P = 0.0011$. $n = 12$ neurons from 4 animals in each group in (**l**, **m**, **o**, **p**). **q** Sample traces of miniature excitatory postsynaptic current (mEPSC). **r** Cumulative probability and comparison of means of mEPSC amplitude. One-way ANOVA, $F_{(3,54)} = 5.125$, $P = 0.0034$. **s** Cumulative probability and means of mEPSC frequency. $F_{(3,54)} = 0.03437$, $P = 0.9914$. **t** Sample traces of miniature inhibitory postsynaptic current (mIPSC). **u** Cumulative probability and comparison of means of mIPSC amplitude. One-way ANOVA, $F_{(3,54)} = 5.330$, $P = 0.0027$. **v** Cumulative probability and means of mIPSC frequency. $F_{(3,54)} = 0.2435$, $P = 0.8656$. $n = 14$ neurons from four animals in each group in (**r**, **s**, **u**, **v**). ns no significant difference; \*$P < 0.05$, \*\*$P < 0.01$, \*\*\*\*$P < 0.0001$. All data were presented as mean ± sem.

specific recording of these cells showed unchanged excitability (Fig. 4o–r).

After confirming the specificity of mPFC-BLA pathway, we further explored if this circuit affected the excitability of postsynaptic neurons in BLA. Using identical CRS and treadmill exercise intervention paradigms, we employed an anterograde transsynaptic strategy via an AAV2/1-Cre vector injection in PrL, plus the expression of Cre-dependent fluorescent proteins in BLA nuclei to labeling cells innervated by mPFC (Fig. 5a, b). The whole-cell patch clamp was then performed to characterize these BLA neurons at the downstream of mPFC. As those of BLA-projecting mPFC cells, the E/I balance of BLA neurons biased toward the hyper-excitation under CRS, as shown by the potentiation of mEPSCs (Fig. 5c–e) and depression of mIPSCs (Fig. 5f–h). On the other hand, exercise training effectively reversed these anomalies via suppressing excitatory currents and potentiating inhibitory currents (Fig. 5c–h). All these modulations did not occur in unstressed mice when receiving treadmill exercise (Fig. 5c–h). In summary, chronic exercise selectively suppressed the activity of mPFC-BLA circuit to confer higher resilience against environmental stress, resulting in the relief of anxiety-like behaviors.

## Discussion

Our work demonstrated the modulation of a cortico-amygdala neural circuit by exercise training in improving stress resilience. Currently available explanations for physical exercise and mental health mainly explored the regulation of brain homeostasis such as neuroinflammation[10,24], cerebrovascular function[25,26] or oxidative stress[9,27]. These changes in brain microenvironment further contribute to the acceleration of adult neurogenesis, leading to the improvement of mental[8,28] or cognitive functions[7,29,30]. Our study, however, focused on a specific mPFC-BLA neural circuit which has been established to modulate innate anxiety behaviors[18,31]. Using ex vivo electrophysiology, we demonstrated that chronic exercise largely attenuated the hyperexcitation of this pathway under CRS, thus relieving anxiety-like behaviors.

The mPFC-BLA circuit has been recognized to be closely related with the innate anxiety status and related behaviors. In particular, CRS tend to shift the E/I balance of this pathway toward the excitation[18], providing the neural substrate for the potentiated input to BLA. Our studies agreed with their reports by showing the higher excitability of both BLA-projecting mPFC neurons and BLA cells innervated by mPFC inputs. BLA is one pivotal center for bidirectional control of anxiety-related

behaviors[32] via its outputs to other nuclei such as central amygdala and ventral hippocampus[33]. The elevated excitability of BLA neurons thus affects the brain network, conferring various behavioral phenotypes to reflect the anxiety status of the animal. Of note, as BLA neurons receive inputs from multiple brain regions other than mPFC, including the thalamus or hippocampal CA1[15], it is possible that other brain nuclei also participate in the alleviation of anxiety disorders via their projections to BLA under exercise paradigm.

In our study, 14-day physical exercise effectively alleviated the hyperexcitation of the mPFC-BLA pathway and recovered the normal E/I balance in both mPFC and BLA nuclei. In addition to modulating membrane excitability, the treadmill exercise may also modify the cellular property of BLA neurons, which have been reported to be related with anxiety-like behaviors. For example, recent progresses have indicated the participation of mitophagy[34], neuroinflammation[35] and synaptic remodeling[16] of BLA cells in association with the pathogenesis of anxiety disorders. These studies further implied the possible involvement of cellular events during chronic exercise training. Our recent work has revealed the change of RNA methylation within mPFC-BLA pathway under the same treadmill exercise paradigm on CRS model[36], providing an alternative molecular explanation for the long-term modification of neural circuit underlying exercise intervention.

It is worth noticed that exercise training selectively inhibited BLA-projecting neurons in mPFC, whilst potentiated non-BLA projecting cells. The divergency of circuitry regulation implied the specific role of exercise in modifying anxiety-related neural pathway and supported the inherent complexity of cortical circuits during emotional processing. Current findings suggest that it is over-simplified to correlate the general neural activity of mPFC with the anxiety or depression status. For example, the selective stimulation of mPFC Drd1 pyramidal neurons exerted anxiolytic effects[37], in contrast to our observation that the inhibition of BLA-projecting cells helped to attenuate anxiety behaviors. The explanations for the divergent roles of mPFC in modulating anxiety behaviors may come from two aspects: (1) **Distinct projection targets**. Recent study has suggested the anxiogenic role of the infralimbic to lateral septum circuit, in contrast to the anxiolytic function of the infralimbic-central amygdala pathway[38]. (2) **The modulation of local interneurons**. GABAergic inputs also modify the excitability of mPFC-projecting neurons, forming an extra layer of neuromodulation. For example, oxytocin receptor-expressing interneurons can inhibit the potentiation of pyramidal cells in mPFC to modulate

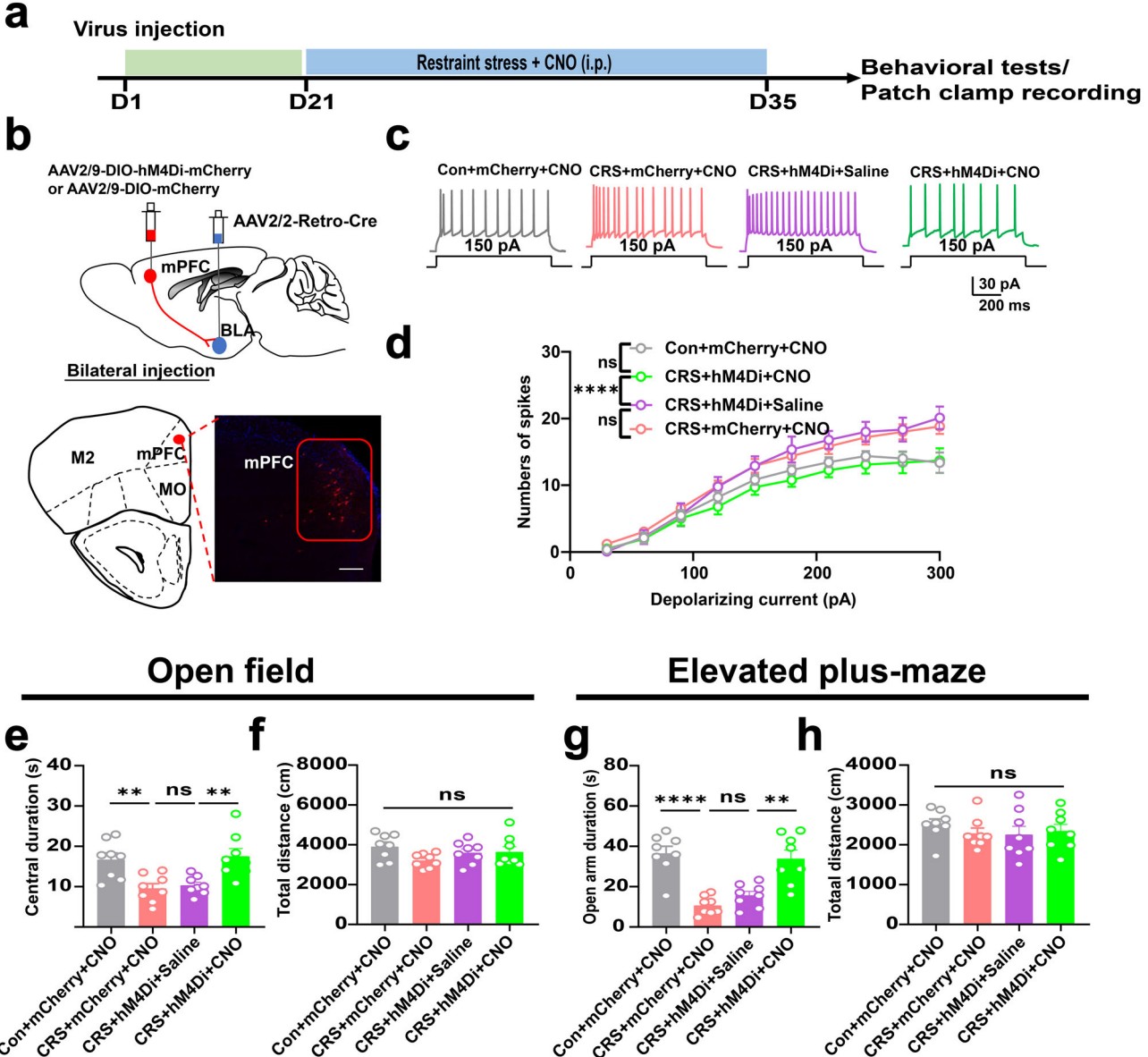

**Fig. 2 Inhibition of mPFC-BLA circuit mimicked exercise effect of anxiolysis. a** Time schedule of chemogenetic manipulation. **b** Upper, experimental scheme of viral injection. Lower, labeling of BLA-projecting neurons in prelimbic region (PrL). Scale bar, 100 μm. **c** Representative traces of mPFC neurons under 150 pA depolarizing currents. **d** Number of spikes under gradient depolarizing currents. Two-way ANOVA with respect to the group effect, $F(3,400) = 21.04$, $P < 0.0001$. $n = 11$ neurons from 4 mice in each group. **e** Total distance in the open field. One-way ANOVA, $F(3,28) = 1.852$, $P = 0.1067$. **f** Central duration in the open field. $F(3,28) = 8.140$, $P = 0.0005$. **g** Total distance in the elevated plus-maze. $F(3,28) = 0.4935$, $P = 0.6897$. **h** Time duration in the open arm. $F(3,28) = 17.82$ $P < 0.0001$. $N = 8$ mice in each group in (**e**–**h**). ns, no significant difference; **$P < 0.01$, ****$P < 0.0001$. All data were presented as mean ± sem.

anxiety behaviors[39], and the suppression of GABAergic signaling in mPFC lead to anxiety-like phenotypes[40]. Such knowledge provides possible identities of those non-BLA projecting cells, which may have distinct distal projecting targets, or belong to local interneurons.

How exercise training modulates the mPFC-BLA circuitry activity remains as one unsolved question in the current study. Possible mechanisms exist in the cellular events within cortical regions, or the reshape of presynaptic inputs to these mPFC neurons. During endurance training, both neurons and glial cells displays dramatic changes, which may profoundly affect the circuitry excitability. Previous knowledge supported the recovery of normal density and morphology of dendritic spines of mPFC under exercise intervention of chronic unpredictable mild stress-

treated rats[41]. Exercise training itself may also modify the oligo-dendrogenesis of mPFC to facilitate the axonal transmission efficiency[42]. Besides the local modulation, physical exercise affects the activity of other brain regions sending inputs to the mPFC. For example, exercise training can reshape the function of hippocampal nuclei[43], which project to mPFC for learning and mental modulations[44]. Moreover, other brain regions related with motor activity, such as the motor cortex, may connect with mPFC to affect its neural activity under exercise paradigms. In future, more sophisticated molecular and circuitry studies are expected to unravel the regulatory network of exercise training.

In addition to the neural circuitry mechanism as stated above, exercise may also modulate different kinds of peripheral factors, which further contribute to the central modulation of neural

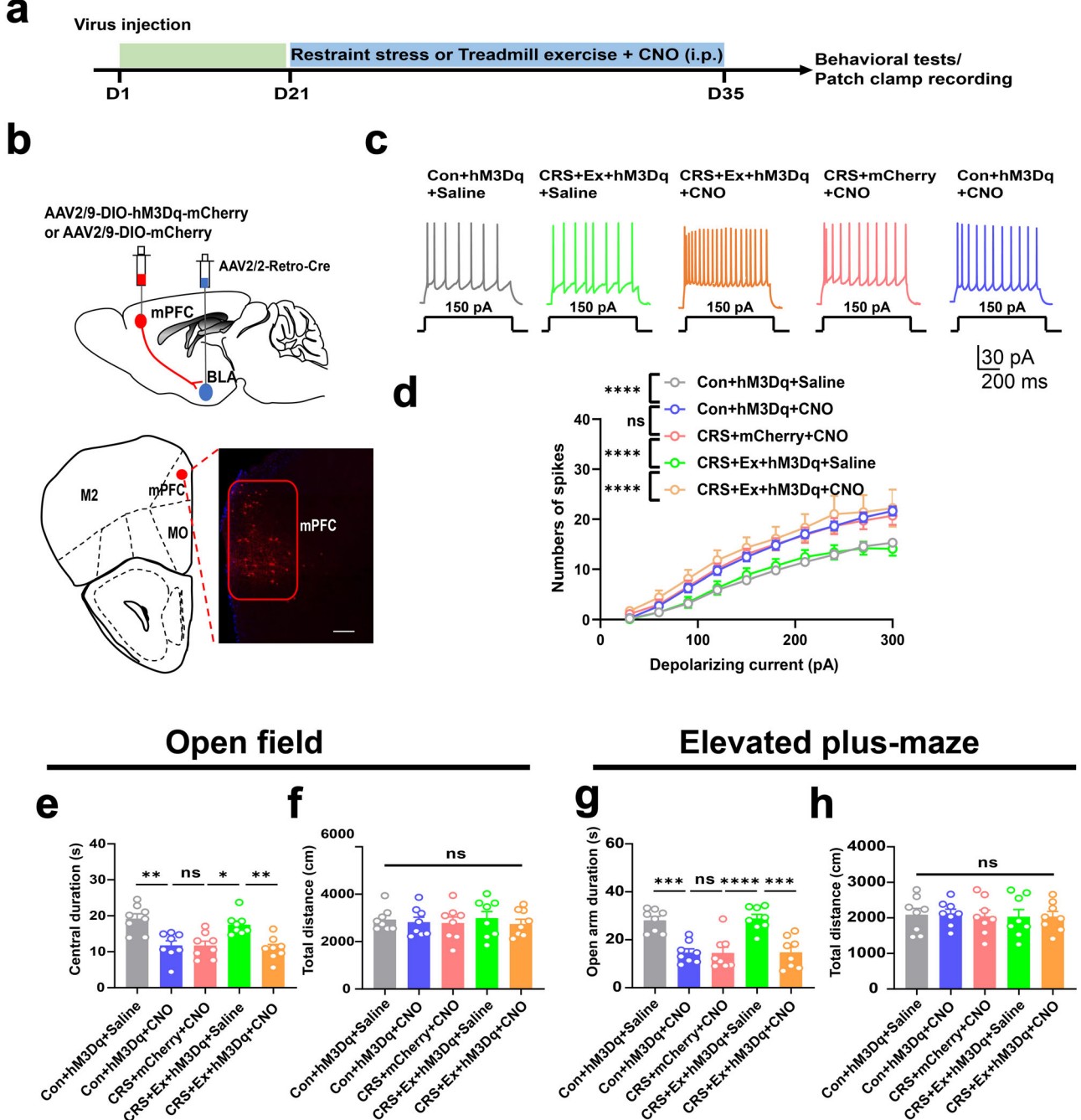

**Fig. 3 Reactivation of mPFC-BLA pathway blocked exercise anxiolytic effect. a** Time schedule of chemogenetic manipulation. **b** Upper, experimental scheme of viral injection. Lower, labeling of BLA-projecting neurons in PrL. Scale bar, 100 μm. **c** Representative traces of PrL neurons under 150 pA injected currents. **d** Number of spikes under gradient depolarizing currents. Two-way ANOVA with respect to the group effect, $F_{(4,520)} = 34.39$, $P < 0.0001$. $n = 11$ neurons from 4 mice in each group. **e** Total distance in the open field. One-way ANOVA, $F_{(4, 35)} = 0,2056$, $P = 0.9336$. **f** Central duration in the open field. $F_{(4,35)} = 9.986$, $P < 0.0008$. **g** Total distance in the elevated plus-maze. $F_{(4,35)} = 0.05699$, $P = 0.9937$. **h** Time duration in the open arm. $F_{(4,35)} = 15.03$, $P < 0.0001$. $N = 8$ mice in each group in (**e**–**h**). ns, no significant difference; *$P < 0.05$; **$P < 0.01$; ***$P < 0.001$; ****$P < 0.0001$. All data were presented as mean ± sem.

network. Our group recently identified a hepatic-secreted metabolite, S-adenosylmethionine (SAM), to be involved in the RNA methylation of synaptic gene transcripts during chronic treadmill exercise for conferring stress resilience[36]. Another adipocyte-derived molecule, clusterin, has also been found to protect cortical neural network via suppressing neuroinflammation[45]. These findings raise the possibility that blood-borne exercise-related factors may interact with specific brain regions to modulate

circuitry activity, forming a more complete molecular-circuitry pathway.

In summary, 14-day treadmill exercise attenuated the hyper-excitation of mPFC-BLA circuit under CRS, leading to the alleviation of anxiety-like behaviors. This study unravels a specific neural circuit to be necessary for exercise-conferred stress resilience and provides novel directions for illustrating the neural mechanism of exercise in preventing mental disorders.

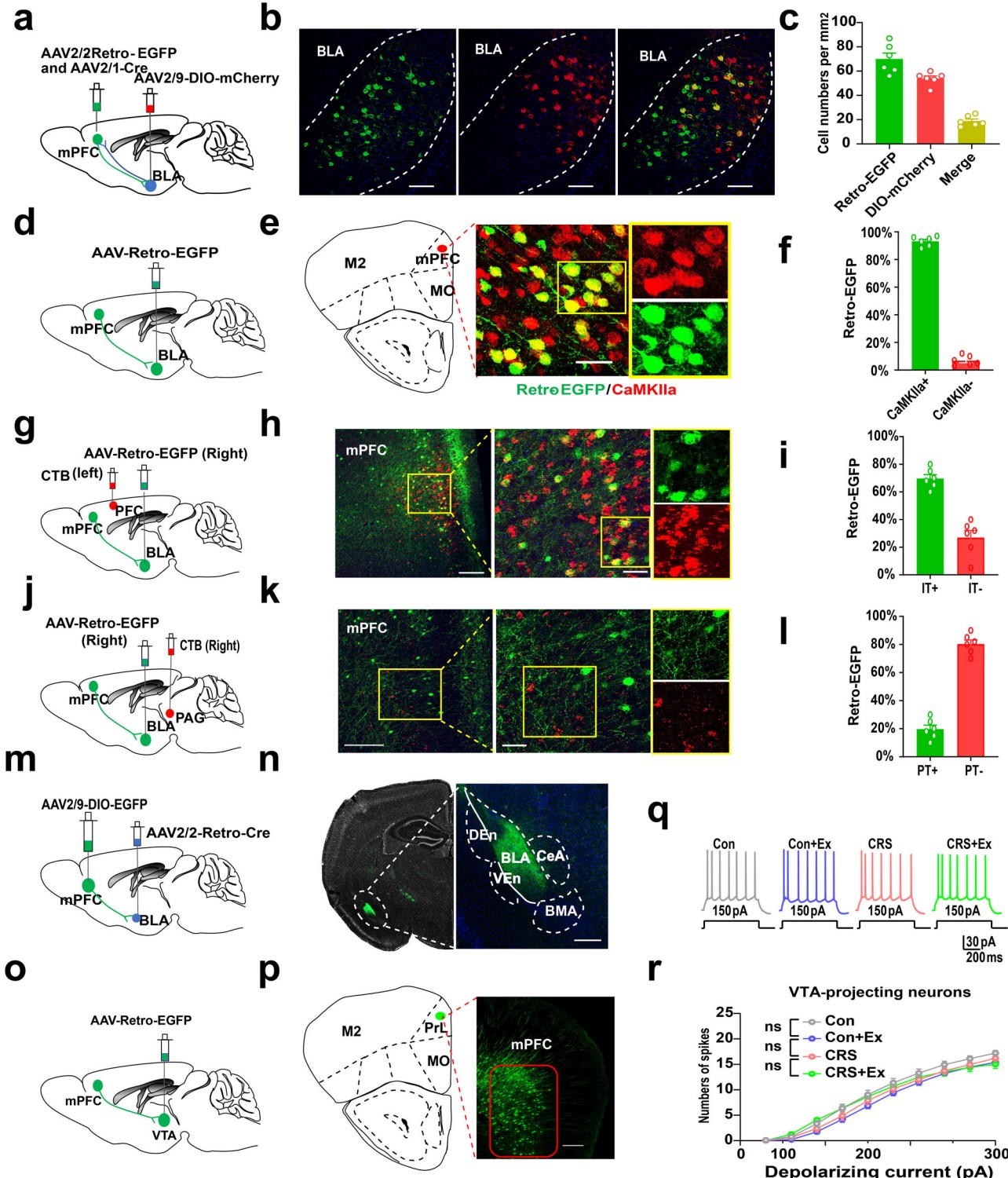

**Fig. 4 Identity of BLA-projecting mPFC neurons. a** Schematic illustration of antero- and retro-grade labeling of mPFC-BLA pathway. **b** Fluorescent images for neuronal tracing. Scale bar, 100 µm. **c** Quantification of antero- and retro-grade labeled cells. **d** Schematics of viral injection. **e** Fluorescent images of BLA-projecting cells and CaMKIIα. **f** Most of BLA-projecting cells expressed CaMKIIα. Two-sample student t-test, $t(10) = 42.18$, $P < 0.0001$. **g** Schematics of viral injection. **h** Fluorescent images of colocalization between BLA-projecting cells and contralateral cortical projecting neurons. Scale bars, 150 µm and 50 µm. **i** More than 70% of BLA-projecting cells belong to intratelencephalic (IT) group. $t(10) = 7.201$, $P < 0.0001$. **j** Schematics of viral injection. **k** Fluorescent images of colocalization between BLA-projecting cells and ipsilateral PAG-projecting neurons. Scale bar, 150 µm and 50 µm. **l** Only a minor fraction of projecting neurons are pyramidal tract (PT) cells. $t(10) = 14.76$, $P < 0.0001$. $N = 6$ mice per group. **m** Schematic diagrams of viral infection. **n** Fluorescence images showing the distribution of axonal terminus of BLA-projecting mPFC cells. Scale bar, 500 µm. **o** Schematic illustration of labeling mPFC-VTA pathway. **p** Fluorescent images showing VTA-projecting mPFC neurons. Scale bar, 100 µm. **q** Samples traces of VTA-projecting mPFC cells given fixed depolarizing currents. **r** No difference of spike number in those neurons after CRS or exercise. Two-way ANOVA with respect to interaction factor, $F(27,620) = 0.6698$, $P = 0.8978$. $n = 11$ neurons from 4 mice in each group. ns, no significant difference. All data were presented as mean ± sem.

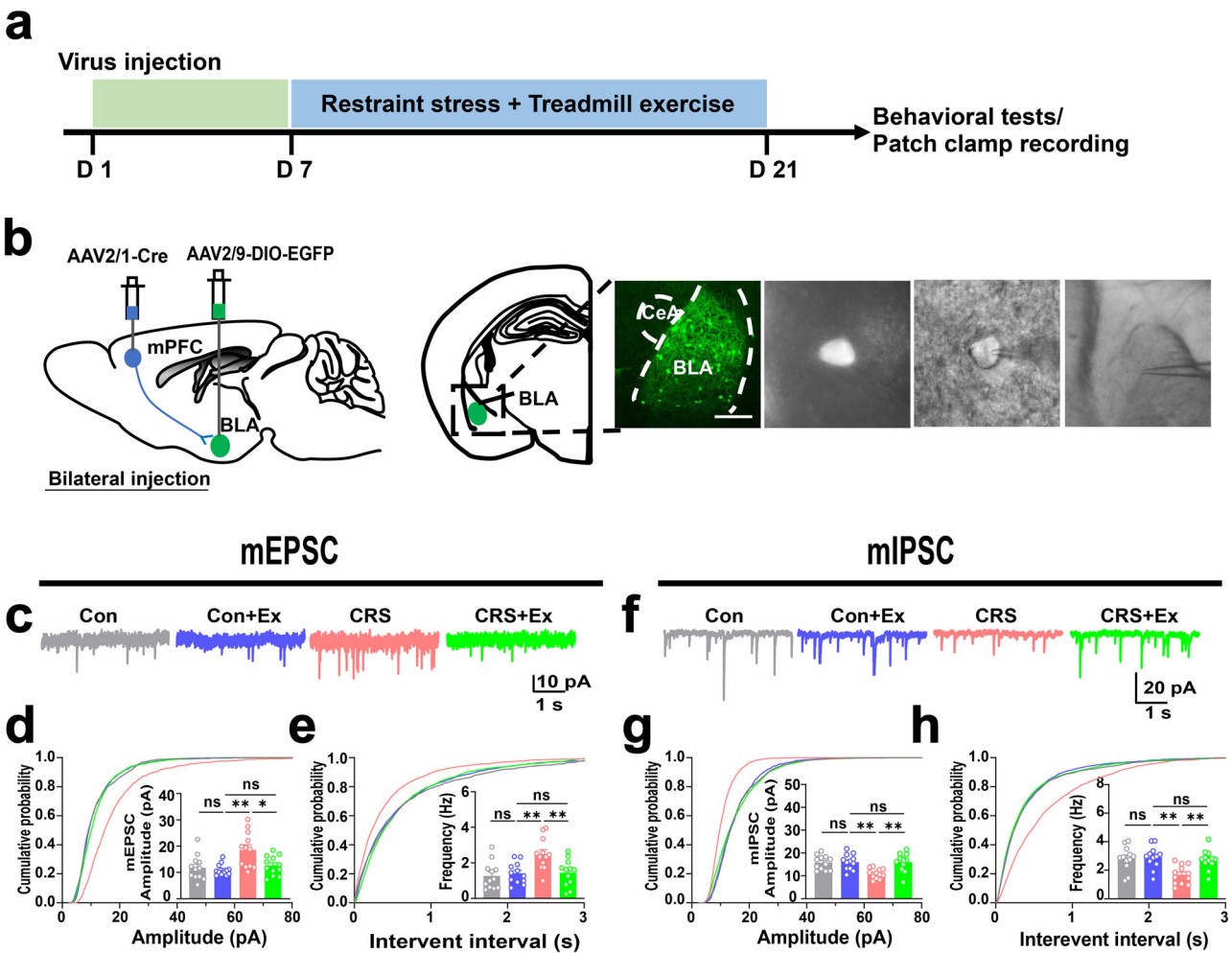

**Fig. 5 Physical exercise reshapes E/I balance of BLA neurons innervated by mPFC. a** Time schedule of experimental design. **b** Left, experimental scheme of viral injection. Right, labeling of BLA neurons innervated by mPFC. Right, fluorescent images (left) and enlarged single neuron for cell attaching record (middle and right). Scale bar, 250 µm. **c** Sample traces of mEPSC of BLA neurons. **d** Cumulative probability and comparison of means of mEPSC amplitude. One-way ANOVA, $F_{(3,44)} = 6.758$, $P = 0.0008$. **e** Cumulative probability and means of mEPSC frequency. $F_{(3,44)} = 6.786$, $P = 0.0007$. $n = 12$ neurons from 4 animals in each group in (**d**, **e**). **f** Sample traces of mIPSC. **g** Cumulative probability and comparison of means of mIPSC amplitude. One-way ANOVA, $F_{(3,44)} = 7.274$, $P = 0.0005$. **h** Cumulative probability and means of mIPSC frequency. $F_{(3,44)} = 7.091$, $P = 0.0005$. $n = 10$ neurons from 4 animals in each group in (**g**, **h**). ns, no significant difference; *$P < 0.05$; **$P < 0.01$. All data were presented as mean ± sem.

## Methods

**Experimental animals**. Male C57BL/6 J mice (4 weeks old) were purchased from Guangdong Medical Laboratory Animal Center. All animals were kept in a standard animal facility with ad libitum access to food and water and humidity (40–60%) and maintained in a temperature (21–25 °C) with a light / dark cycle of 12 h (light on: 8 a.m–8:00 p.m.). Animals were housed at 4 ~ 5 mice per cage. All the experimental procedures were in line with the guidelines of the National Institutes of Health and approved by the Ethics Committee of Experimental Animals of Jinan University.

For chronic stress restraint (CRS) model, mice were placed in a pastry bag for 3 h a day (20:00–23:00) for 14 consecutive days. The pastry bag can wrap the animal very well, which made the animal almost immobile in the pouch. In the exercise model, the mice were put on a treadmill apparatus (Zhongshi Tech., China) for 1 h per day (10:00–11:00), and the velocity was maintained at 10 m/min for 14 consecutive days. A single habituation session was performed at the day before the formal training. The habituation lasted for 10 min, with a velocity setting at 5 m/min. Control mice were placed on a fixed treadmill for 1 h during the same time period.

**Stereotactic injection**. Mice were anesthetized using 1.25% Avertin, and placed in a stereotaxic instrument (RWD, Shenzhen, China). Erythromycin eye ointment was applied to prevent corneal drying. After drilling a hole by the high-speed microdrill, a glass micropipette connected to an ultra-micro injection pump (Nanoliter 2010, WPI, Sarasota, USA) was injected at a slow flow rate of 50 nl/min. The pipette was left in the place for an additional 5 min upon completion of delivery to allow diffusion.

For retrograde tracing the BLA-projecting mPFC neurons, AAV-Retro-EGFP was injected into the BLA (virus titers: $5.4 \times 10^{12}$ GC/ml, 0.12 µL/injection; AP: −1.0 mm, ML: ±3.2 mm, DV: −4.8 mm).

To manipulate the activity of BLA-projecting mPFC neurons specifically for chemogenetically, rAAV2/2-Retro-Cre was injected into the BLA of C57BL/6 mice (virus titers: $3 \times 10^{12}$ GC/ml, 0.1 µL /injection), AAV2/9-DIO-mCherry or AAV2/9-DIO-hM3Dq-mCherry or AAV2/9-DIO-hM4Di-mCherry was injected into the mPFC (virus titers: $3.5 \times 10^{12}$ GC/ml, 0.15 µL/injection; AP: 2.0 mm, ML: ± 0.2 mm, DV: −2.0 mm).

To specifically infect postsynaptic BLA neurons, AAV2/1-Cre (virus titers: $1.5 \times 10^{13}$ GC/ml, 0.1 ml/injection) was injected into the mPFC, AAV2/9-DIO-EGFP was injected into the BLA (virus titers: $3.5 \times 10^{12}$ GC/ml, 0.12 ml/injection).

To determine the BLA-projecting neurons in the mPFC, AAV-Retro-EGFP and CTB were injected in the opposite side of the PFC (AP: 2.0 mm, ML: 0.3 mm, DV: −2.0 mm). Alternatively, AAV-Retro-EGFP and CTB were injected into the mPFC and the PAG (AP: −4 mm, ML: −0.5 mm, DV: −3.0 mm) on the same side.

**Ex vivo electrophysiological recording**. Mice were deeply anesthetized with isoflurane and decapitated, and the brains were quickly removed and coronal slices (250 µm) containing the BLA or the mPFC were cut out by VT1000S Vibratome (Leica Microsystems, Wetzlar, Germany) in ice-cold, oxygenated (95% O2 and 5% CO2) artificial cerebrospinal fluid (ACSF, in mM: 126 NaCl, 2.5 KCl, 1.2 NaH2PO4, 10 Glucose, 26 NaHCO3, 2.4 CaCl2 and 1.2 MgCl2, and 295 mOsm, at pH 7.4). The slices were put into warmed ACSF (33. 5 °C) to recover for 30 min, moved to room temperature and continued to recover for 30 min, and then recorded. Recording electrodes were made from filamented borosilicate glass capillary tubes (inner

diameter, 0.86 μm) by using a horizontal pipette puller (P-97; Sutter Instrument Co., Novato, CA).

To measure the excitability of mPFC neurons, cells were injected with the depolarizing current pulses of 0.8 s with their strength ranging from −90 to 300 pA and increasing in 30 pA steps. The electrodes were filled with K+-based peptide solution (in mM: 135 K-gluconate, 5 KCl, 10 HEPES, 0.2 EGTA, 4 MgATP, 10 Na$_2$-phosphocreatine and 0.3 NaGTP, pH was adjusted to 7.4 with KOH) and the membrane potentials were held at −70 mV.

Miniature excitatory postsynaptic currents (mEPSCs) were recorded in the presence of 1 μM TTX and 20 μM bicuculline, electrodes were filled with K+-based peptide solution. Miniature inhibitory postsynaptic currents (mIPSCs) were recorded in the presence of 1 μM TTX, 20 μM NBQX and 50 μM D-AP5, electrodes were filled with KCl pipette solution (in mM: 140 KCl, 10 HEPES, 0.2 EGTA, 4 MgATP, 10 Na$_2$-phosphocreatine and 0.3 NaGTP, pH was adjusted to 7.4 with KOH). The pipette resistance was around 4–7 MΩ and the membrane potentials were held at −70 mV.

Data were sampled at 10 kHz and Traces were low-pass-filtered at 4 kHz. All recordings were performed using a Multiclamp 700B amplifier (Molecular Devices). Series resistance (Rs) was in the range of 10–20 MΩ and monitored throughout the experiments. If Rs changed >20% during recording, the data were excluded. Offline data analysis was performed using Clampfit 10.0 software (Molecular Devices).

**Open field test**. The open field chamber was made of transparent plastic (50 cm × 50 cm) with an overhead video-tracking system (Noldus, Netherland). Each mouse was gently placed in the center of the chamber and allowed for freely explore the environment for 5 min. During the whole experiment, the time spent in the central area (an arbitrary region of 25 cm × 25 cm) and the total travel of the mice were monitored and analyzed using EthoVision ver 7.0.

**Elevated plus maze**. The elevated plus maze was performed at 24 h after the open field test. The maze is composed of two closed arms (30 cm × 5 cm) and two open arms (30 cm × 5 cm) and a central connecting platform (5 cm × 5 cm), the device is about 74 cm above the ground. Each mouse was placed on the center platform of the maze, facing the open arm, and allowed to explore freely for 5 min. The time spent in the open arms (defined as the mouse has all four paws in the open arm region lasting for >3 s) and the total distance traveled were analyzed using EthoVision ver 7.0.

**Fluorescent immunostaining**. Mice were anesthetized using 1.25% Avertin, saline and 4% paraformaldehyde (PFA) were injected into the heart. The brain tissues were fixed in PFA and were dehydrated in 30% sucrose solution for 48 h. Brain tissue was sectioned into coronal sections containing BLA or mPFC (30 μm thick) using a slide microtome (SM2010R, Leica, Germany). After PBS washing and BSA blocking, the brain slices were incubated with a primary antibody (Rabbit Anti-CaMKIIa, #ab_305050, Abcam; 1:500) at 4 °C for 48 h, followed by a secondary antibody (DyLight 594 Goat anti-rabbit; # AB_36933, Thermo Fisher Scientific, 1:500). Using a confocal microscope (ZEISS, Germany) to capture confocal immunofluorescence images and the fluorescent intensity was analyzed using ImageJ.

**Chemogenetic manipulation**. The mice recovered for 3 weeks after virus injection, and the chronic restraint stress model was established continuously for 14 days. At the same time, CNO (1 mg/kg) or vehicle (0.9% NaCl) was injected intraperitoneally at a fixed time (04:00 p.m.) every day until the day before the behavioral test. Using OPF and EPM to monitor anxiety-like behavior in mice.

**Statistics and reproducibility**. The statistical analyses were performed by GraphPad Prism 7.0 (La Jolla, CA, USA). All data were presented as means ± SEM. Each dataset was first tested for normality, and parametric tests were used if the dataset passed the normality test. Two-sample Student's t-test was used for comparison between two groups, and one-way analysis of variance (ANOVA) was used for differences between multiple groups, followed by Tukey's post-hoc comparison. For two independent variables, two-way ANOVA and Bonferroni post hoc comparison were adopted. Multi-group nonparametric comparisons and two-group nonparametric comparisons were performed with Kruskal–Wallis test and Kolmogorov–Smirnov test, respectively. All the data were analyzed by researchers who were blinded to the experimental conditions. Statistical significance was set at $P < 0.05$. The sample size (number of animals per group) was defined in the figure legend of each experiment. No technical replicate was performed (i.e. each data plot comes from an independent sample).

**Reporting summary**. Further information on research design is available in the Nature Portfolio Reporting Summary linked to this article.

## Data availability

All source data supporting the conclusion of this work have been included in the Supplementary Data file. Other data are available upon request to the corresponding author (zhangli@jnu.edu.cn).

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

## Acknowledgements
This study was funded by STI2030-Major Projects (2022ZD0207600) and National Key Research and Development Program of China (2020YFA0113600) to L.Z., National Natural Science Foundation of China (32070955 to L.Z., and U22A20301 to K.-F.S.), Guangdong Basic and Applied Basic Research Foundation (2023B1515040015) to L.Z., and Science and Technology Program of Guangzhou, China (202007030012) to K.-F.S. and L.Z.

## Author contributions
Z.L. designed and conceived all experiments, with the assistance from J.C. and Y.D., who also participated in the data collection and analysis. The manuscript was prepared by L.Z. and Z.L. with inputs from all authors. This study was supervised by K.S. and L.Z.

## Competing interests
The authors declare no competing interests.
