## [Peer Review File · Communications Biology]

Reviewers' comments:

Reviewer #1 (Remarks to the Author):

In this study, Luo et al investigated the mPFC-related circuit mechanisms underlying exercise-mediated anxiolytic effects. To this end, they made a relatively thorough behavioral and physiological investigations on these circuits during stress and upon exercise training. They found that the chronic restraint stress selectively caused hyperexcitation in BLA-projecting PFC neurons while leaving unaltered those not-projecting to BLA. Moreover, long term treadmill exercise could effectively reverse the phenotypes in PFC-BLA neurons. While this work provides some new information on the circuit mechanism for stress associated anxiety and the prevention of anxiety by exercise, there are multiple fundamental issues to be addressed in detail – see below.

1, Whether restraint stress also alters the excitatory-inhibitory (E/I) balance of these non-BLA projecting mPFC neurons?

2, The authors divided the mPFC neurons into two populations according to their downstream targets (the BLA or elsewhere) and found that restraint stress increased the frequency of spikes in BLA-projecting neurons in mPFC, which were depressed in exercise group. However, no effects were observed on the number of spikes of these non-BLA projecting cells under restraint stress. As generally known, the mPFC neurons projecting other regions other than BLA also get involved in stress-induced anxiety. Thus, no changes on the non-BLA projecting cells by restraint stress only represent the net effect of restraint stress on the PFC neurons projecting multiple ex-PFC regions, it is thus worth testing the effects of restraint stress and exercise on the PFC neurons projecting to specific brain regions other than non-BLA.

3, The mPFC neurons in different layers are largely heterogeneous in terms of their neurophysiological properties, for example, the pyramidal neurons in L2/3 were more hyperpolarized and less excitable than in L5. Thus, whether restraint stress induces different changes between BLA- and non-BLA projecting neurons in L2/3 and L5 layers on electrophysiological signatures in Figure 1 and if so, whether exercise can also reverse?

4, The figures 3 and 4 are actually the same, this mistake is nearly unforgivable. We cannot get any summary information related to the Fig. 4 results in the text. The authors should be more careful and replace with the real Fig. 4.

5, Many ambiguous and inaccurate descriptions throughout the manuscript. For example, line 27, should add "we found...." before " the mPFC-BLA"; Line 169. Should insert "stress-induced" between "prevents" and "anxiety"; Line 208. What does " infecting receptors into neurons" Mean?

Reviewer #2 (Remarks to the Author):

In this study, Luo and collaborators found that mPFC-BLA pathway presented activation in a mouse model of CRS-induced anxiety, and exercise training can reverse this anomaly. In addition, they demonstrated that inhibition of mPFC-BLA pathway was necessary for exercise-mediated anxiolysis. Overall, the experiments are well designed and the conclusion is generally supported by their data. However, following issues should be addressed before it can be accepted for publication.

1, Figure 4 is the same as figure 3, please replace the right one.

2, The authors employed AAV2/1 to label BLA neurons innervated by mPFC. However, one limitation of using AAV1 for anterograde transneuronal studies, is the fact that AAV1-Cre can be also transported in the retrograde direction (Zingg et al. 2016). Application should therefore be limited to pathways that do not contain reciprocal connections between targeted pre- and postsynaptic regions. It's therefore, the recorded neurons in BLA in this study may be the neurons innervated by mPFC or the neuron projecting to mPFC. The authors should rule out the later possibility.

3, The authors used two kinds of anesthetics, isoflurane and Avertin, please give the reasons.

4, line 505, the discrimination "labelling of BLA-projecting neurons in mPFC" did not match the fig5c.

5, line 133, "each mice" should be "each mouse".

6, line 150-151, the information about primary and secondary antibody is missing.

7, line 469 and line 472, "immunofluorescent staining" should be "fluorescent images".

Reviewer #3 (Remarks to the Author):

The authors examine the exercise modulates the medial prefrontal-amygdala neural circuit to improve the resilience against chronic restraint stress. The reviewer also thinks that the topic that the effects of exercise on mood regulation via mPFC-BLA circuit may be of very interest to the field. This manuscript provides an important contribution to the exercise therapy for mental illnesses, and understanding its neural basis. However, the reviewer has minor concerns about the exercise model in this study.

comment

1. What elements of the exercise changed the mPFC-BLA circuit? It is considered that exercise affects the whole body, such as improving cardiorespiratory fitness and glucose or fatty acid metabolism, and increased energy expenditure. Recent studies show that several organs-to-brain cross talk. Therefore, more information could be provided on this, or at least some discussion of this issue provided.

2. The treadmill exercise velocity in this study (10m/min) was relatively low, because 20m/min are defined as "moderate" in normal control rodents from lactate threshold (shima et al., Diabetologia, 2016). Although the reviewer originally considered that low-intensity exercise is effective for CNS, it is concerned about whether the benefits of exercise are actually being achieved. Therefore, other exercise-induced effects should be presented in some way, such as circulating factor, protein or mRNA expression. Nevertheless, even if there were no other exercise effects, this data, which even low-intensity exercise regulates mPFC-BLA circuits, is also interesting and beneficial information for exercise therapy for achieving stress resilience.

3. Under a non-stress situation, similar to the current study, are mPFC-BLA circuits modulated by exercise and contributing anxiolytic effects? It is known that exercise reduces anxiety-like behaviors measured by EPM test in normal rodents as authors cited Tomiga et al., 2021. In Fig.1 and 4, Con+Ex group is missing. Therefore, the current finding, which is the exercise training modulates mPFC-BLA circuit, it is limited in only the CRS model.

4. The author should describe exercise training protocol more precisely, such as habituation period to treadmill apparatus.

5. How to treat control animals during exercise intervention? For instance, in the case of treadmill exercise, non-exercise control animals are placed on a stationary treadmill, for the same amount of time.

6. The authors should describe the number of mice per cage in the Method section, because social isolation or the existence of cage mates were important factors.

7. The authors performed two behavioral tests. However, the order in which these tests were performed is not mentioned. The author should describe that.

8. The authors should explain about behavioral tests more detail. How defined the central area in

OFT? How defined the enter to the open arm (generally, it is defined by entry of all four paws)? This detailed information would be very helpful to the reader.

9. Fig. 4 in the original submitted manuscript looks the same as Fig. 3. Please check it.

Dear Reviewer Experts,

Thank you for all efforts in processing and reviewing our manuscript. All of your suggestions and questions will largely improve the integrity of our work. Now we have finished extra lab works following your guidance and modified the figures and texts (see highlighted sections) as the revised version. Moreover, we have prepared a point-to-point response letter to address each question.

We look forward to your further information.

Li Zhang
Jinan University

****The beginning of the response letter****

Reviewer #1 (Remarks to the Author):

In this study, Luo et al investigated the mPFC-related circuit mechanisms underlying exercise-mediated anxiolytic effects. To this end, they made a relatively thorough behavioral and physiological investigations on these circuits during stress and upon exercise training. They found that the chronic restraint stress selectively caused hyperexcitation in BLA-projecting PFC neurons while leaving unaltered those not-projecting to BLA. Moreover, long term treadmill exercise could effectively reverse the phenotypes in PFC-BLA neurons. While this work provides some new information on the circuit mechanism for stress associated anxiety and the prevention of anxiety by exercise, there are multiple fundamental issues to be addressed in detail – see below.

1, Whether restraint stress also alters the excitatory-inhibitory (E/I) balance of these non-BLA projecting mPFC neurons?

A: Thank you for the valuable advice. We have checked the E/I balance of non-BLA projecting mPFC neurons using a similar approach. Results (see revised Fig. 1q-v) showed the differential modulatory patterns: In short, non-BLA projecting neurons tend to be inhibited under CRS, and are reactivated by exercise.

Fig. R1 Exercise enhances the excitability of non-BLA projecting neurons in mPFC. From revised Fig. 1q-v.

2, The authors divided the mPFC neurons into two populations according to their downstream targets (the BLA or elsewhere) and found that restraint stress increased the frequency of spikes in BLA-projecting neurons in mPFC, which were depressed in exercise group. However, no effects were observed on the number of spikes of these non-BLA projecting cells under restraint stress. As generally known, the mPFC neurons projecting other regions other than BLA also get involved in stress-induced anxiety. Thus, no changes on the non-BLA projecting cells by restraint stress only represent the net effect of restraint stress on the PFC neurons projecting multiple ex-PFC regions, it is thus worth testing the effects of restraint stress and exercise on the PFC neurons projecting to specific brain regions other than non-BLA.

A: We totally agree with your opinions that our work may simply the change of non-BLA projecting mPFC neurons. We have tested the effect of exercise on VTA-projecting PrL neurons using similar approaches (Fig. 4o-r). Results did not reveal significant change of spiking number of these VTA-projecting neurons. Therefore, we believe that exercise-activated, non-BLA projecting PrL neurons (Fig. 1i-j) might belong to corticocortical projecting or corticostriatal projecting neurons.

Fig. R2 VTA-projecting PrL neurons were not affected by CRS or exercise. From revised Fig.

4o-r.

3, The mPFC neurons in different layers are largely heterogeneous in terms of their neurophysiological properties, for example, the pyramidal neurons in L2/3 were more hyperpolarized and less excitable than in L5. Thus, whether restraint stress induces different changes between BLA- and non-BLA projecting neurons in L2/3 and L5 layers on electrophysiological signatures in Figure 1 and if so, whether exercise can also reverse?

A: Thank you for the advice and we apologized for not giving complete information about the patch clamp assay. In our experiments (Fig. 1), only L5 pyramidal neurons were selected. In future we are going to investigate if similar patterns occur in L2/3 projecting neurons.

4, The figures 3 and 4 are actually the same, this mistake is nearly unforgivable. We cannot get any summary information related to the Fig. 4 results in the text. The authors should be more careful and replace with the real Fig. 4.

A: We deeply apologize for this serious mistake. We have uploaded the correct form of Fig. 3 and Fig. 4 (now Fig. 5 in revised version).

5, Many ambiguous and inaccurate descriptions throughout the manuscript. For example, line 27, should add "we found..." before " the mPFC-BLA"; Line 169. Should insert "stress-induced" between "prevents" and "anxiety"; Line 208. What does " infecting receptors into neurons" Mean?

A: Thank you for the advice and we have revised the whole manuscript to correct these issues as well as similar problems.

Reviewer #2 (Remarks to the Author):

In this study, Luo and collaborators found that mPFC-BLA pathway presented activation in a mouse model of CRS-induced anxiety, and exercise training can reverse this anomaly. In addition, they demonstrated that inhibition of mPFC-BLA pathway was necessary for exercise-mediated anxiolysis. Overall, the experiments are well designed and the conclusion is generally supported by their data. However, following issues should be addressed before it can be accepted for publication.

1, Figure 4 is the same as figure 3, please replace the right one.

A: We apologize for this mistake and have uploaded correct form of Fig. 3 and Fig. 4 (now Fig. 5) in the revised version.

2, The authors employed AAV2/1 to label BLA neurons innervated by mPFC. However, one limitation of using AAV1 for anterograde transneuronal studies, is the fact that AAV1-Cre can be also transported in the retrograde direction (Zingg et al. 2016). Application should

therefore be limited to pathways that do not contain reciprocal connections between targeted pre- and postsynaptic regions. It's therefore, the recorded neurons in BLA in this study may be the neurons innervated by mPFC or the neuron projecting to mPFC. The authors should rule out the later possibility.

A: Thank you for the suggestion. To exclude such possibility, we have co-injected AAV2-Retro-GFP and AAV2/1-Cre into PrL, along with injecting AAV2-dio-mCherry into downstream BLA. Quantification showed that less than 20% of total fluorescently labelled BLA neurons express both EGFP (for retrograde) and mCherry (for anterograde from PrL). Other BLA cells only express one fluorescence, meaning that they are either mPFC-projecting, or mPFC-innervated neurons. See revised Fig. 4a-c for details.

Fig. R3 AAV2/1-mediated anterograde labelling is highly specific and with little retrograde labelling cells. From revised Fig. 4a-c.

3, The authors used two kinds of anesthetics, isoflurane and Avertin, please give the reasons.

A: As stated in the Methods section, we used Avertin on living animals, mainly for stereotactic injection surgery as it can rapidly work and maintain for a relatively longer time. For animals sacrificed for patch clamp or brain sampling, we used overdosage of isoflurane in a chamber which can accommodate multiple animals in a time.

4, line 505, the discrimination "labelling of BLA-projecting neurons in mPFC" did not match the fig5c.

A: Thank you and we have fixed this issue.

5, line 133, "each mice" should be "each mouse".

A: Thank you and this typo error (and similar mistakes) has been corrected.

6, line 150-151, the information about primary and secondary antibody is missing.

A: We have added information about antibodies in the revised Methods section (see page 8).

7, line 469 and line 472, "immunofluorescent staining" should be "fluorescent images".

A: We have fixed this issue.

Reviewer #3 (Remarks to the Author):

The authors examine the exercise modulates the medial prefrontal-amygdala neural circuit to improve the resilience against chronic restraint stress. The reviewer also thinks that the topic that the effects of exercise on mood regulation via mPFC-BLA circuit may be of very interest to the field. This manuscript provides an important contribution to the exercise therapy for mental illnesses, and understanding its neural basis. However, the reviewer has minor concerns about the exercise model in this study.

comment

1. What elements of the exercise changed the mPFC-BLA circuit? It is considered that exercise affects the whole body, such as improving cardiorespiratory fitness and glucose or fatty acid metabolism, and increased energy expenditure. Recent studies show that several organs-to-brain cross talk. Therefore, more information could be provided on this, or at least some discussion of this issue provided.

A: Thank you for this suggestion. In addition to the neural circuitry mechanism as stated above, exercise may also modulate different kinds of peripheral factors, which further contribute to the central modulation of neural network. Our group recently identified a hepatic-secreted metabolite, S-adenosylmethionine (SAM), to be involved in the RNA methylation of synaptic gene transcripts during chronic treadmill exercise for conferring stress resilience [*Advanced Science* 2022, PMID 35642952]. Another adipocyte-derived molecule, clusterin, has also been found to protect cortical neural network via suppressing neuroinflammation [*Cell Reports* 2023, PMID 36924491]. These findings raise the possibility that blood-borne exercise-related factors may interact with specific brain regions to modulate circuitry activity, forming a more complete molecular-circuitry pathway. We have added such contents in the Discussion section (page 15-16)

2. The treadmill exercise velocity in this study (10m/min) was relatively low, because 20m/min are defined as "moderate" in normal control rodents from lactate threshold (shima et al., *Diabetologia*, 2016). Although the reviewer originally considered that low-intensity exercise is effective for CNS, it is concerned about whether the benefits of exercise are actually being achieved. Therefore, other exercise-induced effects should be presented in some way, such as circulating factor, protein or mRNA expression. Nevertheless, even if there were no other exercise effects, this data, which even low-intensity exercise regulates mPFC-BLA circuits, is also interesting and beneficial information for exercise therapy for achieving stress resilience.

A: We understand your concerns about the intensity of treadmill. Although velocity as high as 20 m/min has been used in some metabolic studies, a majority of mice studies (also our work) adopted 10-12 m/min for the chronic daily training paradigm (i.e. *Cell Reports*, 2022 Jul 12;40(2):111058; *J Neuroinflammation*. 2022 Oct 4;19(1):243.). We think the 10-12 m/min parameter is more acceptable and has been recognized as a moderate intensity for mice (see *STAR Protoc*. 2021 Feb 5;2(1):100331, which defines ~22 cm/s, or ~12 m/min as "moderate intensity").

For your suggestions about exercise-related circulating factors, we agree with this model and actually has done some works related to this hypothesis. See our response to the above Q1 and Discussion section (page 15-16).

3. Under a non-stress situation, similar to the current study, are mPFC-BLA circuits modulated by exercise and contributing anxiolytic effects? It is known that exercise reduces anxiety-like behaviors measured by EPM test in normal rodents as authors cited Tomiga et al., 2021. In Fig.1 and 4, Con+Ex group is missing. Therefore, the current finding, which is the exercise training modulates mPFC-BLA circuit, it is limited in only the CRS model.

A: Thank you for the suggestion, We have added the Con+Ex group into Fig. 1 and Fig. 5. Results showed that exercise on naïve, unstressed mice did not largely affect the excitability of either BLA-projecting or non-BLA-projecting mPFC cells (Fig. 1). Con+Ex also did not change excitability of PrL-innervated BLA cells (Fig. 5). Therefore, the effect of exercise on mPFC-BLA circuitry excitability is more specific under CRS condition.

4. The author should describe exercise training protocol more precisely, such as habituation period to treadmill apparatus.

A: Thank you for the suggestion. Following established protocols (STAR Protoc. 2021 Feb 5;2(1):100331), a single habituation session was performed at the day before the formal training. The habituation lasted for 10 min, with a velocity setting at 5 m/min. We have added relevant description in Methods section (see page 4).

5. How to treat control animals during exercise intervention? For instance, in the case of treadmill exercise, non-exercise control animals are placed on a stationary treadmill, for the same amount of time.

A: Yes. Control mice were also placed on to a “fixed” treadmill for 1 hr, to minimize the contextual effect. We have added these descriptions in Methods section (see page 4).

6. The authors should describe the number of mice per cage in the Method section, because social isolation or the existence of cage mates were important factors.

A: All mice were housed in 4-5 animals per cage, which is a standard protocol under animal ethical codes. We have added such information in Methods section (see page 4).

7. The authors performed two behavioral tests. However, the order in which these tests were performed is not mentioned. The author should describe that.

A: We firstly performed open field test, followed by the elevated plus-maze (with 24 hr interval). We have added such information in Methods section (see page 7).

8. The authors should explain about behavioral tests more detail. How defined the central area in OFT? How defined the enter to the open arm (generally, it is defined by entry of all four paws)? This detailed information would be very helpful to the reader.

A: Thank you for the reminder. We have added more detailed information about the behavioral test protocols. See Methods section (page 7) for details.

9. Fig. 4 in the original submitted manuscript looks the same as Fig. 3. Please check it.

A: We apologize for this mistake and have uploaded correct form of Fig. 3 and Fig. 4 in the revised version.

REVIEWERS' COMMENTS:

Reviewer #1 (Remarks to the Author):

The authors have addressed many of my concerns. Overall, this study is quite interesting and the results are exciting. However, some minor issues still should be resolved before it can be published.

1, In the figure 1f, the representative image showing layers of mPFC was not precise, as in rodent mPFC, the mPFC is lack of layer 4.

2, In the figure 4h and k, the expanded images seem not right, the authors should check it.

Reviewer #2 (Remarks to the Author):

The authors have addressed my questions.

Reviewer #3 (Remarks to the Author):

The authors have responded appropriately to my concerns, providing additional data.

Dear Reviewer Experts,

Thank you again for all efforts in processing and reviewing our manuscript. We have finished the revision of the figure and text, according to your suggestions. Moreover, we have prepared a point-to-point response letter to address each question.

Li Zhang
Jinan University

****The beginning of the response letter****

Reviewer #1 (Remarks to the Author):

The authors have addressed many of my concerns. Overall, this study is quite interesting and the results are exciting. However, some minor issues still should be resolved before it can be published.

1, In the figure 1f, the representative image showing layers of mPFC was not precise, as in rodent mPFC, the mPFC is lack of layer 4.

A: Thank you for the reminder. We have prepared a new version of Fig 1f to show the layers of mPFC.

Revised Fig. 1f

2, In the figure 4h and k, the expanded images seem not right, the authors should check it.

A: We apologize for this error and have updated the correct image in to Fig. 4h and 4k.

Revised Fig. 4h & 4k